# Performance Analysis of Existing ITS Technologies: Evaluation and Coexistence

**DOI:** 10.3390/s22249570

**Published:** 2022-12-07

**Authors:** Sassi Maaloul, Hasnaa Aniss, Leo Mendiboure, Marion Berbineau

**Affiliations:** 1COSYS-ERENA Lab, University Gustave Eiffel, 33400 Talence, France; 2Computer Sciences & Mathematics Department, JUNIA Engineering School, 59000 Lille, France; 3COSYS-LEOST Lab, University Gustave Eiffel, IFSTTAR, 59650 Villeneuve d’Ascq, France

**Keywords:** vehicular networks, ITS technologies, performance evaluation, co-existence, adaptive-configuration

## Abstract

The performance of vehicular communication technologies changes dynamically according to the application requirements considering data rate, communication ranges, latency, etc. These applications are evolving rapidly and should enhance intelligent transport systems (ITS) such as road safety and automated driving. However, to reach the required quality, these applications need many radio resources to carry the potential traffic load resulting from the environmental perception and data exchanged between the different entities. Therefore, an assessment of vehicular communication technologies’ reliability and resilience under these conditions is required to address the multiple challenges of the ITS services. The paper’s main contribution is to propose a comprehensive analysis model able to evaluate and compare the performances of ITS technologies according to different constraints related to environment-changing situations. This analysis examines the channel occupancy and provides simulation results which allow the identification of the suitable configurations and the most appropriate technology for a given use case. We also propose a coexistence solution between these technologies based on density-sharing according to the use case requirements and the availability of the technology. Finally, we present the challenge of adaptive configuration in vehicular networks, which helps to provide the optimal structure through road profiles and environment variability (infrastructure, data, etc.). Results show different trade offs and limitations between the considered ITS technologies, which are essential to understand their behaviour in a realistic environment.

## 1. Introduction

The Cooperative intelligent transportation systems (C-ITS) have been subject to several research challenges in vehicular networks. The C-ITS services rely on these vehicular networks’ connectivities to enable data exchanges between vehicles and their surroundings. The main goal is to enhance road safety by addressing multiple use cases associated with road safety, traffic efficiency, comfort, etc. In addition, emerging communication technologies drive the development of C-ITS allowing many other advanced use cases related to the connected and autonomous vehicles (CAV) technologies [1,2].

The promising networks that allow these connectivities, specifically vehicle-to-everything (V2X) connectivities, are the ITS technologies LTE -V2X and ITS-G5 as shown in Figure 1. The 3GPP proposes the LTE-V2X or C-V2X (cellular-V2X) solution to extend the LTE standard (in releases 14 and 15), mixing long- and short-range communication. The ITS-G5 was presented by the European Telecommunications Standards Institute (ETSI) based on IEEE 802.11p (2012), which is considered a promising solution for short-range communication technology in C-ITS services.

For reliability, the C-ITS services are deployed in different use cases. Each use case represents a potential vehicular situation with its requirements (e.g., collision warning, traffic information, automated driving, remote driving, etc.). The data exchanged between the different communications entities through ITS technologies are encapsulated in various messages. For example, the road safety and non-safety data are encapsulated in cooperative awareness messages (CAM) and decentralized environmental notification messages (DENM) messages. In addition, other clusters of messages, such as platooning and cooperative perception messages (CPM) are defined to support road infrastructure development and the emergence of the automotive industry. This latter relies on environment perception to build more interactivity with the environment and connected objects to enable further C-ITS services for the benefit of CAV technologies.

These ITS technologies allow vehicles to communicate with diverse entities to enable C-ITS services. For example, the entities in their surroundings would enable vehicles to communicate directly in short-range communications, such as between vehicles (vehicle-to-vehicle (V2V)), between the vehicles and infrastructure (vehicle-to-infrastructure(V2I)) and between the vehicles and other road users such as pedestrians (vehicle-to-pedestrian (V2P)). Similarly, vehicles can disseminate their messages based on the long range of cellular infrastructure in vehicle-to-network (V2N) communications, allowing vehicular users to benefit from many other services, such as non-safety applications and Internet access.

The increasing need for connectivity can significantly affect the performance provided by these ITS technologies, affecting the reliability of C-ITS service, which depends on the use case requirement (e.g., critical warning referred to the road safety event). Moreover, various constraints related to the variability of the environmental context can affect this performance, such as the density and size of data exchanged. In this context, a complete analysis of ITS technology performance is still rarely investigated to address the challenges of the environment-changing situations and better support the diversity of C-ITS services and their requirements.

The main contribution of this study is to propose a comprehensive analysis model that can evaluate ITS technologies according to different parameters, such as data rate, channel capacity and density. Furthermore, this evaluation model’s outcomes depend on the particular requirement of a use case. As such, the following issues are identified:The suitable configurations for each ITS technology according to a specific use case;The pros and cons of ITS technologies are identified (evaluated and compared) to determine their suitability for a use case requirement;A density-sharing proposition for the coexistence between these ITS technologies;Suggest adaptive configuration mechanisms’ interest in system performance, adapting the need to the user context through road profiles and environment variability (infrastructure, data, etc.).

The rest of the paper is organized as follows. Section 2 describes both ITS technologies’ PHY/MAC layers and illustrates an overview of the previous evaluation work. Section 3 evaluates theoretical channel occupancy and provides simulation results for performance evaluation and comparison. The new studies on spectrum-sharing methodologies between ITS Technologies are summarised in Section 4 with simulation results for the coexistence proposition. The challenge of the feasibility of adapting the network configuration to environment-changing situations is addressed in Section 5. Finally, Section 6 concludes the paper and suggests some future directions.

## 2. Fundamentals and Related Work

This section describes mechanisms and protocols at the physical (PHY) and medium access control (MAC) layers used by ITS-G5 and LTE-V2X standards. Then, a summary of previous evaluations of ITS technologies is presented, and issues that need to be addressed are discussed.

### 2.1. Its-G5 MAC/PHY Layer

The current IEEE 802.11p was standardized in 2009 as an extension of IEEE 802.11a (Wi-Fi standard), which is expected to support V2X communications due to outside the context of BSS (OCB) mode. This latter adopts OFDM at the PHY layer and the carrier sense multiple access protocols with collision avoidance (CSMA/CA) protocol at the MAC layer to manage the access to the communication medium. The OFDM PHY divides the available frequency spectrum 10 MHz used for the vehicular environment at 5.9 GHz into narrower subchannels (156.25 kHz). Thus, the high-rate data stream is split into various lower-rate data streams transmitted simultaneously over several subcarriers, where each subcarrier is narrow-banded. There are 52 subcarriers, where 48 are used for data, and 4 are pilot carriers. The OFDM PHY layer supports eight different transfer rates, achieved using various modulation schemes and coding (MCS) rates, as shown in Table 1. All transmission is mandatorily transmitted with a coding rate of 1/2, which corresponds to a transfer rate of 3 Mbit/s, 6 Mbit/s and 12 Mbit/s [3]. Therefore, the duration of one OFDM symbol is 8 µs, and in each symbol, different data bits can be carried, depending on the modulation scheme and coding. Therefore, the packet duration depends on the selected transfer rate and the packet’s length. For example, a message duration of 350 bytes (without overhead) with MCS 2 requires 0.466 ms to be transmitted.

The algorithm deployed by 802.11p to allow transmission at the MAC layer is called enhanced distributed coordination access (EDCA). This one is based on the primary distributed coordination function (DCF), a CSMA/CA algorithm and added QoS attributes. ITS-G5 is an asynchronous ad hoc protocol that starts listening to the channel before transmission for a predetermined listening period and transmits directly if the channel is perceived as idle. Otherwise, the station will perform a back-off procedure that defers its access to a randomized period. The EDCA mechanism defines four access categories (ACs) that provide support for the delivery of traffic.

### 2.2. C-V2x MAC/PHY Layer

#### 2.2.1. LTE-V2X

LTE-V2X [4] is a synchronous network that can operate in the 5.9 GHz band with 10 MHz or 20 MHz channels for the vehicular environment. The LTE-V2X is based on single carrier frequency division multiple access (SC-FDMA), with a flexible resource allocation in the time–frequency domain. The resource is divided into subframes fixed to 1 ms in the time domain. In the frequency domain, the channel bandwidth is divided into subchannels with multiple resource blocks (RBs) of 180 kHz. Each RB is composed of 12 subcarriers separated by 15 kHz. Every subcarrier transmits 14 OFDMA symbols per subframe (9 symbols for data transmission, 4 symbols for demodulation reference signals (DMRS) and the last symbol to switch transmission/reception). For each transport block (TB), depending on the packet length, one or more subchannels are attributed for a fixed transmission time Interval (TTI) equal to 1 ms. The transmission of TB requires a control information SCI (sidelink control information) that occupies 2 RBs and helps the correct reception and demodulation of the TB. The resource allocation for the TB and their associated SCI are placed on adjacent or non-adjacent blocks, which must be transmitted in the same TTI.

The communication mode between vehicles in coverage conditions stays on the base station (cellular-assisted) to manage radio resource allocation for direct communication via the PC5 interface. Otherwise, LTE-V2X is based on pre-configured parameters for a sensing mechanism before transmission to attribute radio resources outside network coverage. However, one of the proposed approaches for sensing is based on the semi-persistent scheduling (SPS) mechanism. This mechanism uses various parameters defined at the PHY and MAC layers, referred to as channel sensing, resource allocation, reservation periods, etc. The vehicle needs to sense the transmission channel and identify potential resources. First, the available resources are selected in an interval [T1, T2]. These resources should be below a given threshold of measured power level, where (1 ms ≦ T1 ≦ 4 ms) and (20 ms ≦ T2 ≦ 100 ms) present, respectively, the first and the last TTI of the selection window. Then, a small portion of those resources is usually considered the candidates’ resources (the recommended value is around 20%). Next, the vehicle randomly selects candidate resources for subsequent transmission among all received candidate resources. This resource is used for a random re-selection counter in the range [5, 15], decremented at each communication. When this latter reaches zero, the vehicle can keep the same resource with a probability Pk (keeping probability) or repeat the resource selection process with probability (1-Pk).

#### 2.2.2. 5G-V2X

Several enhancements to V2X communications have been made by 3GPP [5,6] within the 5G system and its new radio (NR) known as 5G-V2X or NR-V2X. It is defined to support requirements of advanced V2X use cases such as advanced driving applications. In the case of LTE, the OFDM’s subcarrier spacing (SCS) has a single value of 15 kHz. Two slots form the sub-frame, and the TTI has a fixed value of 1 ms that corresponds to the duration of the sub-frame.

For the NR-V2X sidelink, the slot duration and their number in the subframe depend on the SCS of the OFDM. The SCS can be obtained with different OFDM numerologies as shown in Table 2, which is flexible for NR V2X. Thus, a larger SCS results in a shorter slot duration as shown in Figure 2. A slot consists of 12 or 14 OFDM symbols depending on the cyclic prefix (normal CP or an extended CP). Therefore, the NR-5G provides flexibility in resource allocation per slot where the number of OFDM symbols depends on the OFDM numerologies and can be flexible (UL or DL direction).

NR V2X defines two modes (Modes 1 and 2) for sidelink resource allocation. In Mode 1, referred to as Mode 3 in LTE-V2X, radio resources for V2V communications are controlled by the network. In Mode 2, referred to as Mode 4 in LTE-V2X, UEs can directly select their sidelink resources through an autonomous selection using a channel sensing mechanism. The resources are specified within the resource pool and referred to as contiguous or non-contiguous available physical resource blocks (PRBs) for transmission.

5G NR differs from LTE-V2X through more flexible spectrum management. For example, PHY layer numerology achieves flexibility, guaranteeing lower latency with high numerologies. In addition, the MAC layer supports more variety of data traffic (periodic and non-aperiodic). Compared to LTE-V2X, NR V2X introduce sidelink retransmissions through blind retransmissions or HARQ-feedback (hybrid automatic repeat request) and supports broadcast, group cast and unicast communications.

### 2.3. Previous Evaluation Work

The assessment of ITS technologies (ITS-G5 and LTE-V2X) has been addressed in previous studies to enable vehicular networks. The approaches were mainly oriented as follows:We have found several research studies on vehicular networks that address and evaluate the actual performance of vehicular communications technologies. These studies are related to analysis and compare the suitability of ITS technologies to support different performance keys [7,8,9,10,11,12,13], such as congestion situation, data rate, reception ratio, latency, etc.In addition, mechanisms and protocols used by each technology have been studied separately to propose the most appropriate configuration. In this context, the studies [14,15,16,17,18,19] focus on the performance analysis of resource allocation parameters (at the PHY/MAC layers) considered by the SPS procedure in LTE-V2X technology. These studies were conducted under different scenarios to evaluate the effect of various parameters, such as selection and sensing window, transmit power, probability and reselection counter, etc. In addition, other works [20,21,22,23,24] propose performance evaluation of the direct communications using IEEE 802.11p (dedicated short-range communications (DSRC) or ITS-G5) to support application requirements under diverse situations such as communications range, data rate, congestion, etc.

Therefore, the literature does not present a complete study that evaluates these ITS technologies’ boundaries and capabilities. We consider that the assessment study remains open for the following reasons:The diversification of use cases and their requirements;The variability of vehicular density with road capacity;The amount and kind of data exchanged according to the road environment perception;The limitation on simulation tools.

However, this work intends to provide a comprehensive and complementary study to previous research. The objective is to establish a general evaluation model that simulates each technology’s performance boundaries according to several constraints, such as density, range, PRR and delay. This approach identifies suitable configurations and the most appropriate technology for a given use case. The operating constraints of these use cases depend and vary based on the road infrastructure capacity and data exchanged. The aim is to assess ITS technologies according to their suitability to use case requirements. In addition, based on this model, we propose a co-existence solution between ITS technologies based on density-sharing for the required use case. Although this paper assumes that both technologies operate simultaneously in different frequency bands, their PHY/MAC layers parameters are not modified. Therefore, the channel selection mechanism is out of the scope of this present work.

## 3. Performance Evaluation and Comparision

This section reviews the channel occupancy in each ITS technology and examines their different characteristics and problems, and the second part of this section provides a complementary simulation analysis that evaluates and compares the performance supplied by both ITS technologies.

### 3.1. Theoretical Evaluation of Channel Occupancy

The channel occupancy for a data packet with ITS-G5 and LTE-V2X is theoretically compared. However, as mentioned above, the ITS-G5 is an asynchronous network that allows channel allocation to one user at a time (no users-multiplexing). Therefore, the occupancy time depends on the message size as well as the transfer rate. For example, a data packet of 350 bytes (with 40 μs for overhead) requires 0.973 ms with MCS 0, 0.506 ms with MCS 2 and 0.273 with MCS 4 to be transmitted.

In contrast, LTE-V2X is a synchronous network that allows multiple users to transmit (users-multiplexing possible in frequency domain) but with message duration fixed to 1 ms. In Table 3, different transport block (TB) sizes and coding rates are tabulated for 10 MHz frequency channels according to [25]. As the table shows, the number of RBs needed to transmit a message varies according to the message size and the coding schemes. The coding rates suggested by standards recommendation [26] are approximately around 0.5 for BPSK and 0.5 or 0.7 for QPSK modulation. For example, a data packet of 350 bytes with MCS 6 requires 28 RBs for TB and 2 RBs for control information SCI. Depending on the subchannel configuration, only one message can be transmitted during a TTI (1 ms) since a 10 MHz channel provides only 50 RBs. Using the same packet length, we can improve the transmission for two messages if we consider MCS 9 and three messages with MCS 13 in the same TTI.

As we can see, the relevant number of transmitted messages is based on the packet size and the configured MCS. First, if we consider the ideal configuration, the performance of both ITS technologies can be similar in terms of transmissions number in the same period of channel occupation. However, suppose the number of RBs of a data packet exceeds 26. Only one message is sent during the entire TTI transmission, and the performance decreases, considering that the remaining resources are unused. Then, the access time to the channel depends on the access technique itself, where LTE-V2X requires additional time for resource allocation according to the selection window before transmitting. For ITS-G5, access is managed by the back-off algorithm if the channel is busy. In this case, the ITS-G5 depends on the efficiency of the back-off procedure. Finally, the resource re-selection counter and the probability of re-using the resource allow keeping the same resource for several transmissions, introducing more performance in specific scenarios (dense situations). However, the ITS-G5 requests channel access at each communication.

The effect of these different examinations, such as channel access delay, message size, access techniques performances and optimal coding scheme, is this paper’s primary focus through evaluating and comparing ITS technologies under the same conditions.

### 3.2. Simulation Parameters and Scenario Description

The LTEV2Vsim [27,28] is considered to study resource allocation for V2V communications in vehicular networks. It was developed based on MATLAB simulator and used to evaluate the performance of realistic vehicular scenarios. LTEV2Vsim defines resource allocation in controlled and autonomous mode for LTE-V2V networks and allows simulating the cooperative awareness service using IEEE 802.11p/ITS-G5. In this work, we evaluate the LTE 4 mode against the ITS-G5 mode, where resources are selected dynamically, as this mode offers stringent performance, especially for autonomous vehicles. According to the respective standards, we chose the slow highway scenario [26]. The scenario corresponds to approximately 2 km of six lanes (three in each direction). All vehicles randomly consider packet generation between 0 and 100 ms, with a medium speed of 50 km/h (standard deviation 5 km/h). This evaluation study only evaluates ITS technologies in environment change conditions, such as network density, packet size and channel capacities. For these conditions, the values used for message size are 350, 800 and 1200 bytes derived from the related work, such as [29,30]. These values are picked to cover the different C-ITS messages (CAM, DENM and CPM). The total system communication density level has values from 50 to 500 vehicles, contributing to the increased channel load and eventual congestion. According to the standards recommendation, a suitable MCS is selected for each technology based on the packet length (see resource allocation section). Table 4 summarises the relevant simulation parameters. Bandwidth is 10 MHz, and simulation time of 150 (s), large enough to repeat the resource selection procedure several times and reach a congestion situation. For MAC/PHY layers, we consider recommended configurations described and deduced in previous work for both technologies to avoid replication of results.

A packet reception ratio (PRR) of more than 90% was considered to illustrate the density vs. range illustration, used to identify each technology’s capabilities and boundaries. Furthermore, the time constraints presented by the update delay (UD) and date age (DA) are evaluated for time-critical use cases, which, respectively, represents the delay between successively received messages and the lifetime of messages.

### 3.3. Performance Evaluation and Comparision

The main goal of the first part is to define the optimal configurations for each ITS connectivity according to several parameters. These include parameters that constantly vary in environments, such as density and the quantity of data transmitted, and other parameters related to adjusting the channel capacity to the flow, such as the MCS coding. The results obtained allow us to observe the effect of these parameters on the system’s performance, showing the impact of the density on the range and delays.

#### 3.3.1. Performance of ITS-G5 Technology

In this subsection, the ITS-G5 is evaluated for different packet lengths to determine their performance effects and identify the most efficient configurations. The channel configurations considered are provided by MCS 0, MCS 2 and MCS 4, corresponding to a transfer rate of 3 Mbit/s, 6 Mbit/s and 12 Mbit/s.

ITS-G5 adopts the carrier sense multiple access with collision avoidance (CSMA-CA) protocol at the MAC layer to manage access to the communication medium. However, the CSMA/CA protocol showed a significantly high probability of collisions. In this context, ITS-G5 addresses congestion situations through the distributed congestion control (DCC) mechanism [31] to mitigate the risk of collisions. The DCC has been introduced for channel control and adjusts traffic loads dynamically by reducing exchanged information.

It operates based on a DCC management that contains all information provided by DCC access (related to channel occupancy) and the controlled parameters to adjust traffic load. The DCC can adapt the traffic load through a set of algorithms, such as transmit power control (TPC), transmit ratio control (TRC) and transmit data rate control (TDC). As a result, the DCC can better support access management in high-traffic load situations using these algorithms. In Figure 3, we present the delay between generating the message by vehicles and its effective transmission. This delay reflects the time needed to channel access and transmit the packet. As shown in Figure 3, the packet delay increases according to the packet lengths and the density, with better performances for MCS 4, which offers the highest transfer rate. Depending on the channel capacity, the ITS-G5 controls the traffic by relying on the DCC to reduce the network load. This is visible for large packets, such as 1200 bytes, since it requires more time on the channel for transmission. This reduces the number of messages sent and keeps the QoS (transfer delay) to a minimum in ultra-dense areas. Finally, the configuration with MCS 4 provides a transfer rate of 12 Mbit/s, which explains a higher traffic absorption capacity than MCS 2 and MCS 0.

Once we add this duration to the update delay, we find the lifetime of the message (date age), while the update delay is the time between successively received messages. For ITS-G5, the packet delay is practically negligible compared to the update delay. As a result, the update delay and date age are almost similar in the following simulations.

In a scenario of small message exchange (corresponding to the size of CAM message), as shown in Figure 4, the MCS 4 coding provides more performance in terms of range for high densities (≥220 veh/km), as well as a delay (UD et DA) for 95% of the received messages. This is explained by the channel’s ability to handle many messages in a dense environment. On the other hand, for low densities below 220 veh/km, MCS 2 coding becomes more efficient in terms of range (it can reach 200m with a PRR above 90%) and provides similar performance to MCS 4 coding in terms of delay.

In Figure 5, the message size has increased to 800 bytes (corresponding to the amount of data carried by a DENM message). Therefore, the configuration with MCS 2 coding has the highest range for almost all scenarios with different densities. On the other hand, the channel capacity with MCS 4 keeps the best performance in terms of delay, especially for densities above 100 veh/km.

The messages with a large size of 1200 bytes, which are suitable for carrying perception data, such as CPM messages, are shown in Figure 6. Again, the optimal configuration is provided almost by MCS 4 coding, ensuring the best range and reduced update delays for 95% of all received messages.

The packet length can directly affect the performance of ITS-G5 technology, where the optimal configuration with MCS 4 coding providing the highest data rate may not always guarantee the most high performance. Thus, a configuration trade off between range and delay arises in some situations.

#### 3.3.2. Performance Evaluation of LTE-V2X Technology

In this subsection, LTE-V2X is evaluated according to the different message sizes to determine their performance effects and the ideal configurations. In contrast to ITS-G5, the resource allocation for LTE-V2X technology is based on sub-channel allocation. Depending on the TB size, this latter changes with varying the MCS.

Figure 7 defines the update delay and the date age for low and high density. As mentioned in Section 2, LTE-V2X relies on the SPS procedure for resource allocation, where resource selection is made in the selection window (between T1 and T2). Otherwise, the selection is made in the next windows selection in case of resource unavailability. The update delay is almost equal to the transmission frequency (100 ms) in low density. However, the lack of resource availability in the increasingly dense scenario causes new resource reservations in the next selection window, extending the update delay by 100 ms. By adding the packet delay to the update delay, we will effectively have the date age of the message. In Figure 7, the packet delay is the time needed to reach the reserved resource (Tresource allocation) and its transmission. For 95% of all received messages at low density, the date age can reach 195 ms (100 ms for the update delay and 95 ms the time needed for 95% of transmissions). As soon as the density increases, some new resource reservations extend to the next selection window. As shown in Figure 7, the DA evolves with the density from date age min to date age max in the period [UD+T1, UD+T2]. The evolution of the date age allows us to determine the resource occupancy rate for any given density.

For a small packet size of 350 bytes, the possible configurations are provided by MCS 6, MCS 9 and MCS 13 coding, which respect the coding rate relative to the standard recommendation. As shown in Figure 8, MCS 9 provides the best performance on the range and delays for densities above 120 veh/km. On the other hand, MCS 6 offers more coverage than MCS 9 in low densities, reaching more than 220 m (with a PRR above 90%). In addition, this coding provides similar performance as MCS 9 in terms of delay for 95% of total received messages. The occupancy rate of the resources is also visible by date age, where it approaches the update delay for a new reservation in the following selection window; then, it increases.

In Figure 9, the packet size has increased to 800 bytes. The possible configurations for this TB size are provided by the MCS 9 and MCS 13 coding that respect the recommended coding rate. The MCS 9 coding has the highest range for all density scenarios. On the other hand, there is a slight delay in performance with MCS 13 on densities below 350 veh/km.

The CPM messages supporting large data (1200 bytes) are shown in Figure 10. The possible configuration for this TB size is provided practically by the MCS 13 coding. For this large packet size and a PRR of more than 90%, LTE-V2X can support a density of 300 veh/km with a range of 40 m. The delay increases linearly for 95% of received messages as a density function.

MCS coding is directly related to the TB size, where increasing the message size makes it impossible to use small MSCs, considering that the LTE-V2X channel is limited to 50 RBs/10 MHz. Similar to ITS-G5, a configuration trade off between range and delay arises in some situations.

#### 3.3.3. Comparison of ITS Technologies: LTE-V2X/ITS-G5

In this subsection, the results obtained previously have been used to determine the most appropriate configuration for each technology according to the different constraints (message size, density, coding). Then, based on these configurations, we compare them under the same conditions to identify their suitability to a given use case and if one technology outperforms the other.

In the following, we consider the same simulation conditions for the different sizes of messages. Figure 11 shows the optimal configurations for a small message (packet size of 350 bytes) representing a CAM message. The comparison results focus on the range with PRR more than 90% and the update delay for 95% of all received messages. Three parts are identified:The first part considers that the ITS-G5 (with MCS 4 coding) has the best performance (in the areas with densities above 220 veh/km). This result converges with the delay constraint where update delay remains better with the same MCS coding;The second part provides the best performance by LTE-V2X (MCS 9) and ITS-G5 (MCS 2). However, the results are pretty close in range and delay for densities around 100 to 200 veh/km;LTE-V2X (MCS 6) performs better in low-density areas (less than 100 veh/km) in the third part. Therefore, the coverage is favoured for this area while ensuring that the PRR is higher than 90% and has a low update delay.

Contrary to previous results, the comparison between ITS-G5 and LTE-V2X for a medium packet size of 800 bytes supporting DENM messages shows trade offs between range and update delay. In this context, two approaches have been identified:The first approach is range-oriented, which supports use cases that demand strict performances in high density;The second approach is delay-oriented, which minimises the update delay for C-ITS applications with high delay requirements.

According to the first approach, Figure 12 shows a better performance for ITS-G5 (MCS 2) in terms of range on higher densities (above 200 veh/km), against the superiority of LTE-V2X (MCS 9) for lower densities (below 200 veh/km) which can reach 200 m. On the other hand, these MCSs present high updates delays. However, for a scenario requiring more strict delay requirements, as presented in Figure 12, the second approach reduces the update delay for each technology with MCS 4 for ITS-G5 and MCS 13 for LTE-V2X against considerable attenuations in terms of range.

The configuration with high MCS increases the channel capacity to support more data. Therefore, ITS-G5 with MCS 4 and LTE with MCS 13 show the best performance for data with a significant message size of 1200 bytes. Figure 13 shows that ITS-G5 overcomes LTE-V2X for high densities (more than 200/250 veh/km) against an update delay favouring LTE-V2X and vice versa for low densities.

The obtained results allow assessing the behaviour of both technologies under the same conditions. LTE-V2X shows a better range at low density than ITS-G5. Still, ITS-G5 offers a very high capacity to support increased traffic load in high-density scenarios while ensuring that the PRR is always above 90%. The update delay is represented for 95% of all received messages, where it increases not only with density but also with packet sizes. Although the ITS-G5 has a high capacity due to the MCS 4 coding, which provides the best update delay, LTE-V2X also performs well for large messages in dense scenarios due to its SPS procedure. According to the obtained results, there is a trade off between technologies depending on the density level.

## 4. Coexistence of ITS Technologies

This section presents the spectrum band dedicated to vehicular networks and channel coexistence problems between ITS technologies. Subsequently, we define a coexistence solution based on the evaluation model and the results obtained in the previous section.

### 4.1. Spectrum Sharing for ITS Technologies

LTE-V2X and IEEE 802.11p/ITS-G5 wireless standards currently operate in the 5.9 GHz spectrum band to provide short-range communications for C-ITS applications. However, the European CEPT (Conference of Postal and Telecommunications Administrations) in [32] and the ECC (Electronic Communications Committee) decision [33] propose a frequency arrangement. It is allocated on block sizes of 10 MHz to ensure efficient frequency band use, as depicted in Figure 14. According to this arrangement, the safety communications related to road ITS applications shall prioritise access spectrum in the frequency range 5875–5915 MHz. Furthermore, the priority is allowed for safety related to urban rail ITS applications in 5915–5935 MHz. In addition, ITS road I2V applications can access the frequency band 5915–5925 MHz to address the urban rail ITS applications.

The frequency band 5875–5935 MHz is divided into block sizes of 10 MHz, then assigned to ITS technologies based on the channel availability. However, sharing mechanisms are required to manage the channel’s access to ITS technologies in the same geographical area. According to the ETSI standard [34], the preliminary study on spectrum sharing mechanisms has been introduced to define channel priority between ITS technologies for road ITS applications. These mechanisms provide various sharing options that demand modifying the PHY/MAC layers (if necessary), introducing channel access priority, detecting other technology occupying the channel, reducing co-channel interference, etc.

Because of their access channel methods, LTE-V2X and ITS-G5 cannot co-exist in their current states to operate in the same frequency channel. However, we can note that LTE-V2X ensures users multiplexing over various subchannels for fixed TTI of 1 ms against one access per user with ITS-G5 for time duration related to packet lengths and MCS. Thus, radio resource allocation cannot be possible simultaneously. Moreover, the last OFDM symbol with a duration of 71.4 μs (RX/TX switch, sometimes not used) in LTE-V2X allows CSMA/CA to sense channel availability in some potential situations, which causes collision problems. In this direction, the ETSI standard [35] proposes studies on the feasibility of the co-existence between these ITS technologies in the same frequency channel. These studies are based on sharing access in the time domain (time-division multiplexing) to overcome the collision and imbalance in channel access (excessive occupation by one technology).

Currently, the ETSI studies endorse the assumption of simultaneous operation and deployment of both technologies. This assumption allows vehicular networks to support better growth of C-ITS applications (e.g., collective perception, platooning, CAV technologies). Furthermore, consider the benefits of ITS technologies according to use cases’ suitability (e.g., ITS-G5 or LTE-V2X for important CAM/DENM safety messages, cooperative perception messages, high density, range, high PRR, etc.). Hence, the secondary purpose of the paper focuses on providing a complementary study based on performance and capabilities offered by both technologies under various constraints, such as network density and message diversity. This complementary study proposes a co-existence solution based on density sharing between these technologies that helps enhance performance.

### 4.2. Coexistence Based on Density Sharing

Different scenarios for deploying C-ITS services have been studied and implemented worldwide through several projects. In other words, to avoid limiting the dissemination of these services to one technology depending on the country or the car manufacturer, there is a high probability that both technologies will operate simultaneously through multi-interface equipment. Therefore, co-existence is necessary to address these different services not only in performance level but also in terms of mass (density) and accessibility.

In this coexistence study, the proposed solution exploits the benefits and drawbacks identified in the previous analysis. Thus, we consider that each ITS technology operates in a separate frequency band to reduce co-channel interference and support more density. Furthermore, considering that each vehicle uses two interfaces simultaneously, the proposal establishes a density sharing between ITS technologies. This sharing depends on the performance provided by these technologies in density defined by the capacity of the road infrastructure. The coexistence solution can be adjusted according to these approaches:Approach 1: The first approach aims to maximise the range for such scenarios: essential CAM/DENM safety messages, etc.Approach 2: The second approach reduces the time constraints (update delay) addressing specific use cases (e.g., platooning messages, CAV services) requirements.

The first approach has been considered for a use case consisting of a highway scenario with three lanes in each direction and a speed of 140 km/h, as recommended by the standard. Under this scenario, the density can reach a density of 150 veh/km. The results established in the previous section show that the best distribution between ITS technologies is to serve 100 vehicles by LTE-V2X and 50 vehicles by ITS-G5. The proposed approach is compared to LTE-V2X/ITS-G5 coexistence without sharing for different message sizes. In Figure 15, we provide the variation of packets received by the two technologies as a range function. For a PRR exceeding 90%, the first approach shows considerable range improvements (above 50 m) for the different message sizes.

In Figure 16, the density has been increased to 250 veh/km for a large packet size of 1200 bytes. The two approaches have been compared to a solution without sharing. The results show that both approaches have better performance. On the one hand, the density sharing provided by the first approach increases the range (by more than 40 m) against a significant update delay (200 ms for 95% of all received messages). On the other hand, the density sharing for the second approach allows maintaining a relatively minor update delay of around 100 ms for 95% of the received messages against a range attenuation. As the previous section explains, there is always a trade off between the coverage and delay requirements, as shown by the two approaches.

The coexistence and the simultaneous use of both ITS technologies improve the efficiency of vehicular networks significantly. This study demonstrates the feasibility of density distribution between ITS technologies based on contextual changes, such as road infrastructure capacity and data size, related to CAM/DENM/CPM messages.

## 5. Adaptative Configuration

This section addresses a significant challenge in the vehicular network: adapting the network configuration according to environment-changing situations. The idea is to provide guidelines to better support C-ITS services’ diversity and their requirements.

As mentioned in the previous sections, the different constraints and trade offs make it challenging to define an optimal solution with a single configuration. For example, in a real-life environment, the density of vehicles varies according to the capacity of the road infrastructure (e.g., urban, rural and highway), which allows us to predict and limit certain constraints such as density. In addition, data dissemination is ensured, on the one hand, by periodic messages between vehicles and the other entities and event messages triggered by road events.

In this regard, we can differentiate two situations: The first one is periodic steady traffic configured with only one MCS coding depending on the message type (e.g., CAM or CPM). The second situation is combined traffic formed by different messages (e.g., CAM, DENM and CPM), where the optimal configuration depends on the message size. As shown in Table 5, the ideal configuration of these messages depends on the amount of data to be transmitted. In this table, the best coding to achieve the highest PRR (above 90%) has been presented. If there are multiple configurations for the same scenario, we have considered the one that provides the fastest update delay. Otherwise, we maintain them in case of performance equality.

Figure 17 shows an example of mixed traffic (combined with different messages CAM, DENM and CPM), served by the ITS-G5 at several density levels. All messages are simultaneously configured by the same coding (MCS 2 or MCS 4). As shown in Figure 17, MCS 4 offers higher performance (in terms of PRR (above 90%) and range), especially in dense areas. This improvement is due to the ability of the channel configured with MCS 4 coding to absorb the combined traffic. However, as shown in Table 5, it is possible to have scenarios with simple traffic that require different configurations than MCS 4.

Therefore, depending on the scenario, an optimal configuration trade off occurs when switching from combined traffic configured by this MCS 4 coding to other traffic operating with different optimal configurations.

With the massive IoT deployment in road infrastructure, the amount of data exchanged between the different entities grows and changes from one location to another (e.g., urban, rural and highway). To better meet these environment-changing situations, it is necessary to adapt the configuration according to the environment’s needs (density) and traffic variability. This adaptive configuration can be initiated as shown in the sequence diagram in Figure 18:Either by a reconfiguration request following an event triggered by a vehicle, the request is encapsulated in a DENM message and broadcast directly to the other vehicles and the infrastructure: the RSU (road side unit) for ITS-G5 and the cellular network (C-ITS server) for LTE-V2X;Or by the platform road operator (Pfro) on the areas providing data perception. In this case, the request is encapsulated in a CPM message. Then, the road operator initiates the reconfiguration request to the concerned RSUs and the cellular network using the Datex II coding, translating it into a C-ITS message.

This study aims to show interest in defining a suitable configuration. The adaptive configuration request can be managed by the road operator and adapted to a specific context according to road area profiles. This allows a more useful adaptation of the ideal configurations. Furthermore, the profiles can be considered beside the adaptive design. It can also provide information about better distribution (density sharing) between ITS technologies. For example, it is possible to address ITS-G5 in high density for a given message type and LTE-V2X for more coverage with other message types (CAM, DENM).

## 6. Conclusions and Future Work

The high demands of C-ITS services for data exchange require efficient resource management and a procedure to ensure the appropriate performance. This paper provides an analysis model of LTE-V2X and IEEE 802.11p/ITS-G5, which allows evaluating and comparing their boundaries in environment change situations, such as network density, packet size and channel capacities. In addition, based on obtained results, we propose a coexistence solution that relies on the density sharing between ITS technologies exploiting their benefits to maximise system performances. Furthermore, we identify the optimal configurations for each ITS technology to show that resilient systems via an adaptive arrangement are required to adapt to environment-changing situations. All the above findings show different trade offs and limitations between the considered ITS technologies. These conclusions are essential to better understand their behaviour in a realistic environment.

The recommendation on the adaptive reconfiguration concept works effectively as an excellent metric to limit such trade offs. However, many research efforts are needed to improve this concept, which can be further combined with density-sharing distribution to define road profiles. In addition, the new 5G generation has introduced more flexibility in managing resource allocation (e.g., numerologies and MAC layer aspects) [36]. However, these enhancements will affect the performance of V2X applications. Therefore, evaluating this technology under the same conditions is needed for advancing C-ITS use cases.

## Figures and Tables

**Figure 1 sensors-22-09570-f001:**
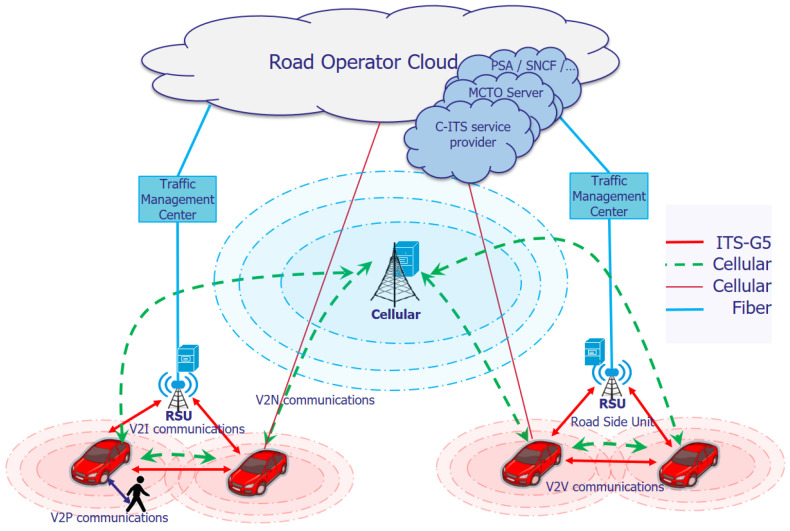
Vehicular communications.

**Figure 2 sensors-22-09570-f002:**
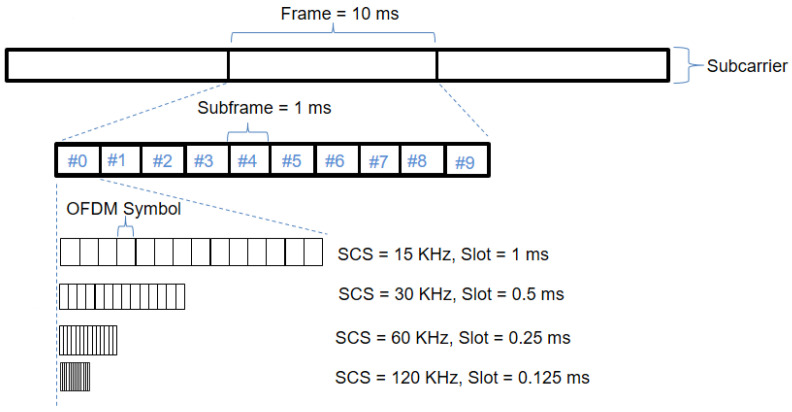
NR-V2X time–frequency structure.

**Figure 3 sensors-22-09570-f003:**
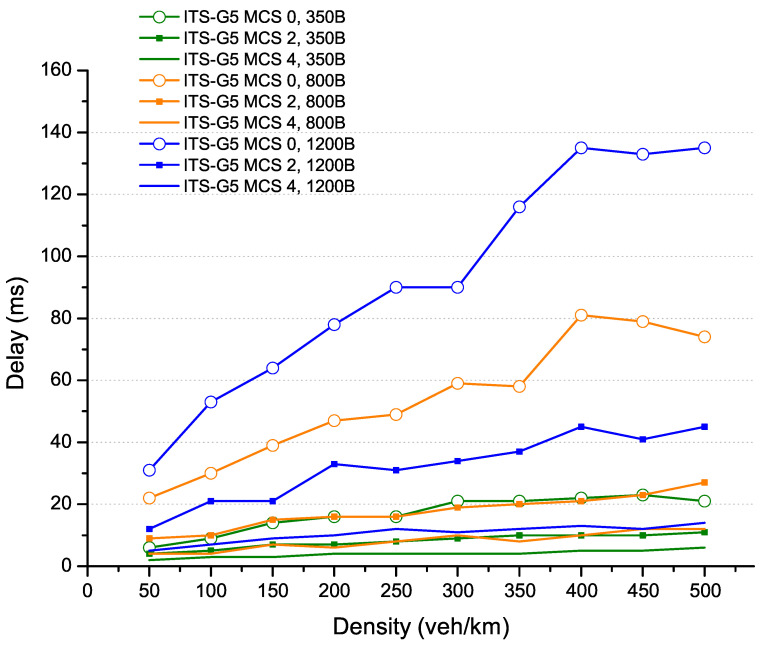
ITS-G5 packet delay for 100% of the generated packet.

**Figure 4 sensors-22-09570-f004:**
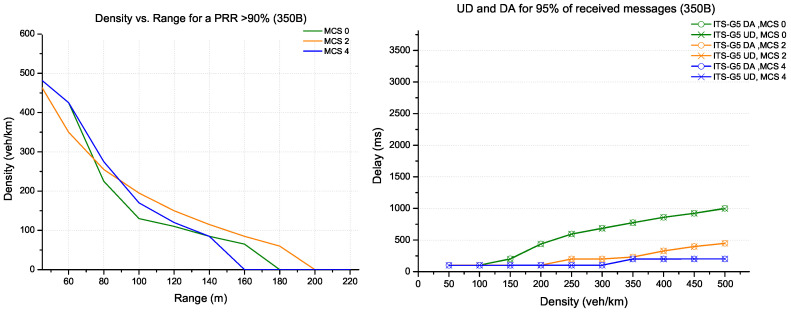
Evaluation for ITS-G5 technology (packet size 350 bytes).

**Figure 5 sensors-22-09570-f005:**
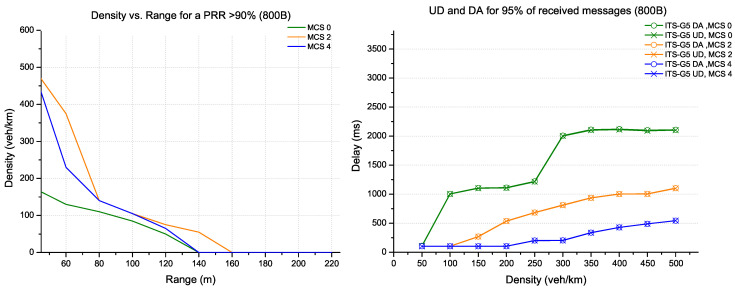
Evaluation for ITS-G5 technology (packet size 800 bytes).

**Figure 6 sensors-22-09570-f006:**
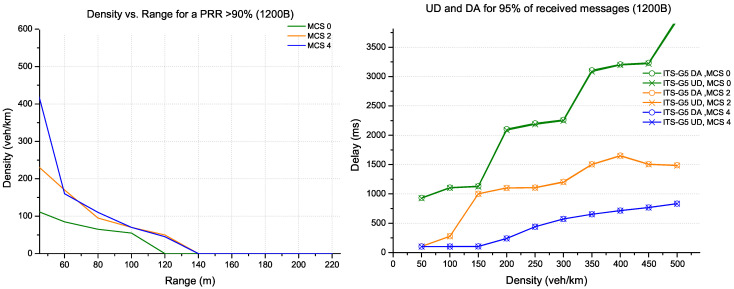
Evaluation for ITS-G5 technology (packet size 1200 bytes).

**Figure 7 sensors-22-09570-f007:**
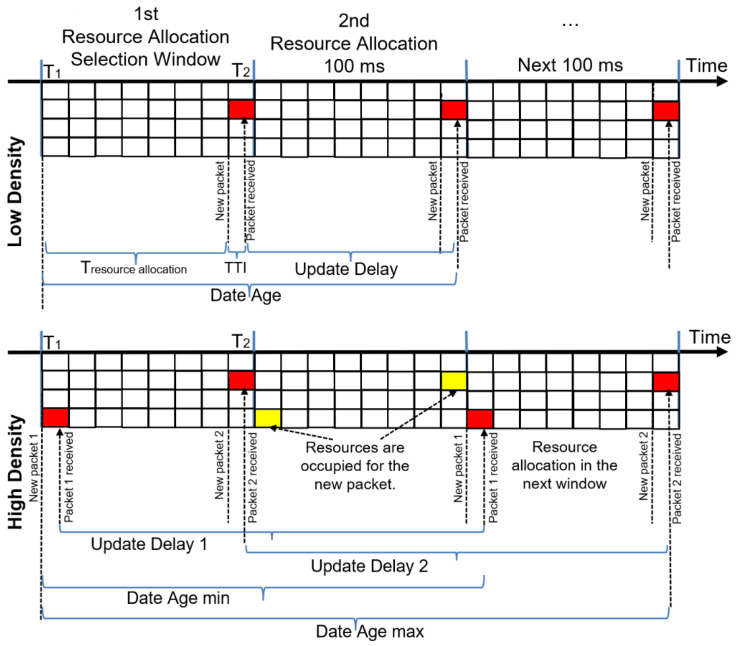
Update delay and date age for LTE-V2X.

**Figure 8 sensors-22-09570-f008:**
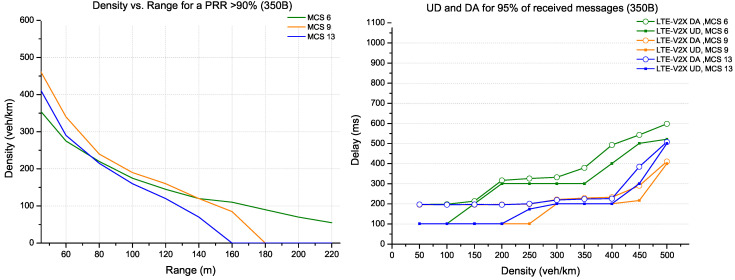
Evaluation for LTE-V2X technology (packet size 350 bytes).

**Figure 9 sensors-22-09570-f009:**
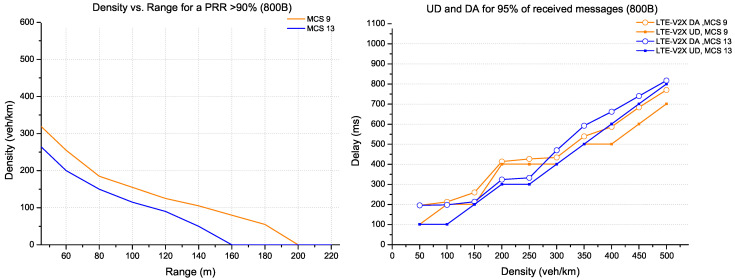
Evaluation for LTE-V2X technology (packet size 800 bytes).

**Figure 10 sensors-22-09570-f010:**
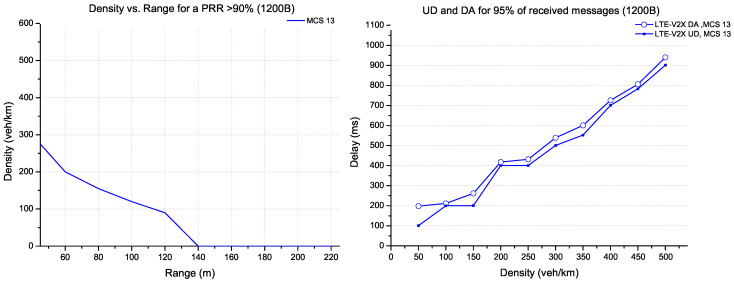
Evaluation for LTE-V2X technology (packet size 1200 bytes).

**Figure 11 sensors-22-09570-f011:**
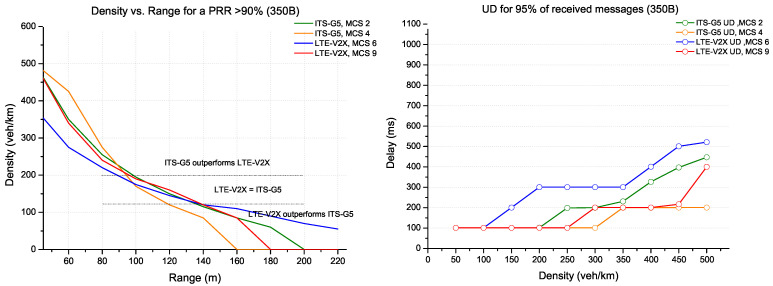
ITS-G5 vs. LTE-V2X comparison for packet size 350 bytes.

**Figure 12 sensors-22-09570-f012:**
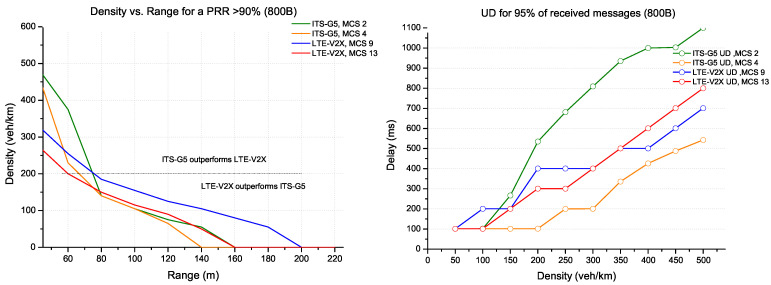
ITS-G5 vs. LTE-V2X comparison for packet size 800 bytes.

**Figure 13 sensors-22-09570-f013:**
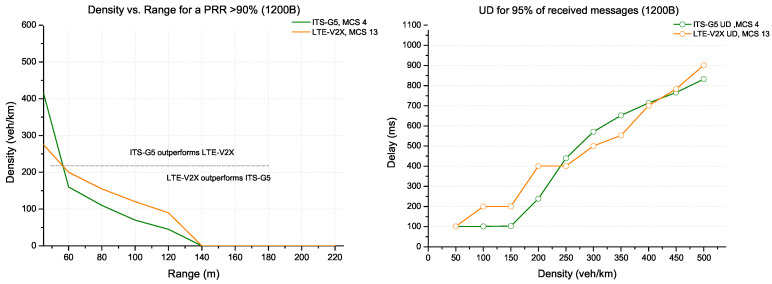
ITS-G5 vs. LTE-V2X comparison for packet size 1200 bytes.

**Figure 14 sensors-22-09570-f014:**
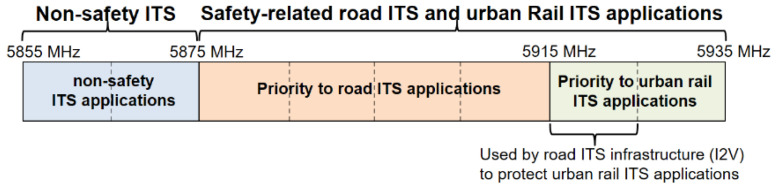
Frequency band for ITS technologies.

**Figure 15 sensors-22-09570-f015:**
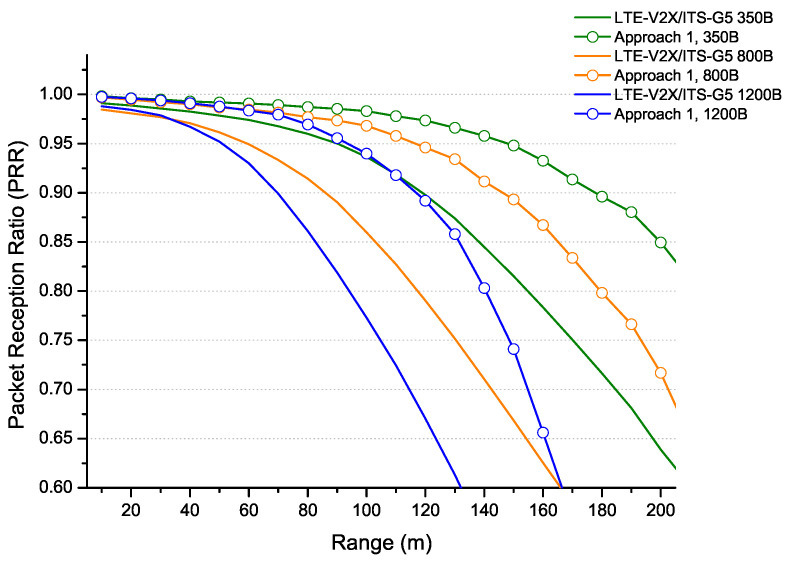
Coexistence LTE-V2X/ITS-G5: density sharing for different packet sizes.

**Figure 16 sensors-22-09570-f016:**
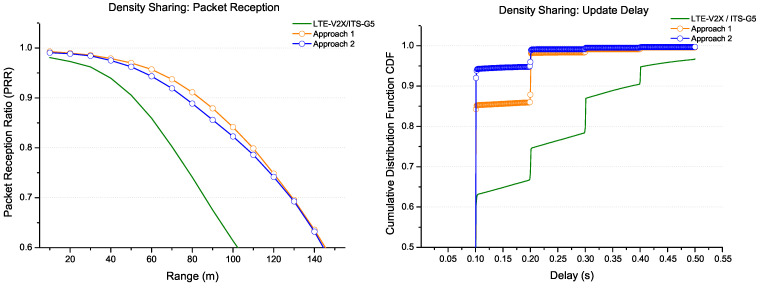
Coexistence LTE-V2X/ITS-G5: density sharing for packet size 1200 B.

**Figure 17 sensors-22-09570-f017:**
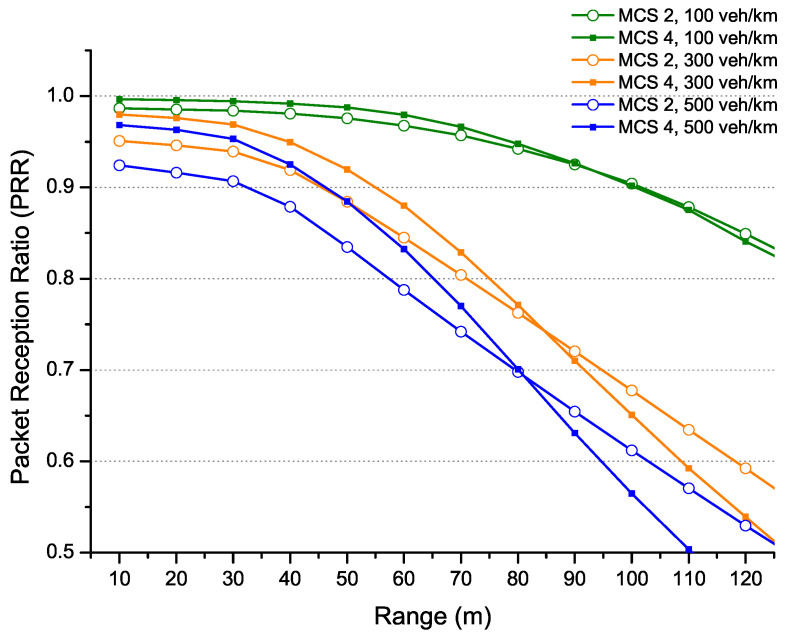
Example of mixed traffic (combined with CAM, DENM, CPM) for ITS-G5.

**Figure 18 sensors-22-09570-f018:**
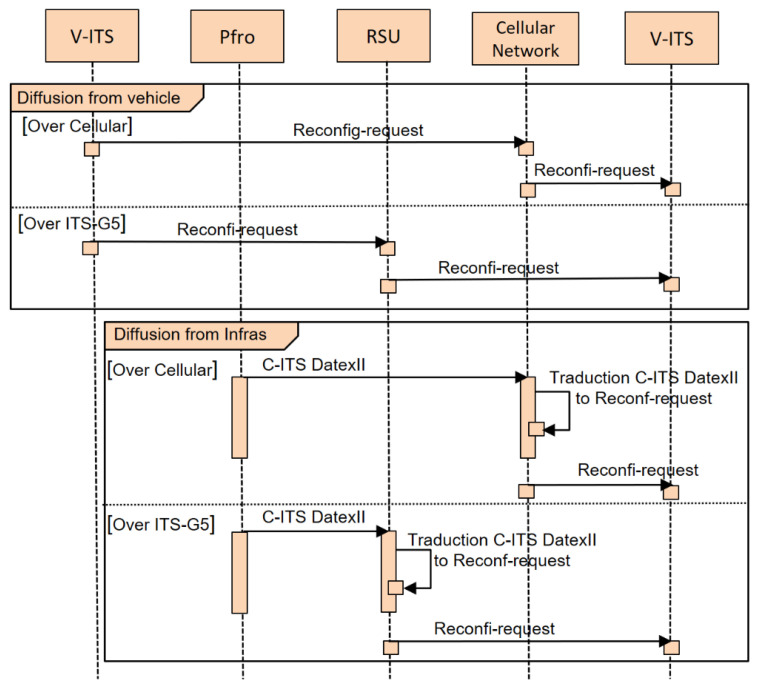
Sequence diagram for reconfiguration diffusion from a vehicle and infrastructure.

**Table 1 sensors-22-09570-t001:** Transfer rates, modulation schemes and coding rates in 802.11p [3].

MCS	Transfer Rate [Mbit/s]	Modulation Scheme	Coding Rate	Data Bits per OFDM Symbol	Coded Bits per OFDM Symbol
MCS0	3	BPSK	1/2	24	48
MCS1	4.5	BPSK	3/4	36	48
MCS2	6	QPSK	1/2	48	96
MCS3	9	QPSK	3/4	72	96
MCS4	12	16-QAM	1/2	96	192
MCS5	18	16-QAM	3/4	144	192
MCS6	24	64-QAM	1/2	192	288
MCS7	27	64-QAM	3/4	216	288

**Table 2 sensors-22-09570-t002:** Supported numerologies in NR-V2X.

Numerology	SCS	Slots per Subframe	Slots per Frame	TTI
0	15 KHz	1	10	1 ms
1	30 KHz	2	20	0.5 ms
2	60 KHz	4	40	0.25 ms
3	120 KHz	8	80	0.125 ms
4	240 KHz	61	160	0.0625 ms

**Table 3 sensors-22-09570-t003:** Coding rate and RBs required for different Transport Block (TB) sizes.

MCS Index	TBS Index	Nb RBs TB = 350 Bytes	Coding Rate	Nb RBs TB = 800 Bytes	Coding Rate	Nb RBs TB = 1200 Bytes	Coding Rate
0	0	-	-	-	-	-	-
1	1	-	-	-	-	-	-
2	2	-	-	-	-	-	-
3	3	-	-	-	-	-	-
4	4	40	0.33333333	-	-	-	-
5	5	33	0.4040404	-	-	-	-
6	6	28	0.4973545	-	-	-	-
7	7	24	0.58024691	-	-	-	-
8	8	21	0.66313933	46	0.65217391	-	-
9	9	18	0.74074074	41	0.73170732	-	-
10	9	18	0.37037037	41	0.36585366	-	-
11	0	17	0.40958606	37	0.40540541	-	-
12	11	15	0.46419753	32	0.46875	-	-
13	12	13	0.53561254	28	0.53571429	43	0.53488372
14	13	11	0.60606061	25	0.6	38	0.60526316
15	14	10	0.66666667	23	0.65217391	34	0.67647059
16	15	10	0.72592593	21	0.71428571	32	0.71875
17	15	10	0.48395062	21	0.47619048	32	0.47916667
18	16	9	0.51577503	20	0.5	30	0.51111111
19	17	8	0.55555556	18	0.55555556	27	0.56790123
20	18	8	0.60493827	17	0.61147422	25	0.61333333

**Table 4 sensors-22-09570-t004:** Simulation parameters.

Simulation Settings	
Scenario	Highway [3GPP recommendation]
Number of lanes	3 in each direction
Road length	2000 m
Vehicle density	50–500 vehicles/km
Bandwidth	10 MHz
Transmission power	23 dBm
Antenna gain (tx and rx)	3 dB
Noise figure	6 dB
Propagation model	WINNER+, Scenario B1
Simulation Time (s)	150
Speed	50 km/h
Speed variation	±5 km/h
Packet size (Bytes)	350, 800, 1200
Periodicity	10 Hz
**ITS-G5**	
Modulation and Coding Scheme	MCS0, MCS2, MCS4
AIFSN	6
Contention Window	15
**LTE-V2X Mode 4**	
Modulation and Coding Scheme	MCS6, MCS9, MCS13
Keep probability	0.8
Resources selection Rsel	0.2 (20%)
Sensing threshold	−110 dBm
Allocation interval [T1, T2]	T1 = 1 ms; T2 = 100 ms
Sensing period	1 s

**Table 5 sensors-22-09570-t005:** Optimal coding rate for PRR ≥ 90%.

Density	Packet Size 350 Bytes		Packet Size 800 Bytes		Packet Size 1200 Bytes	
(Veh/km)	ITS-G5	LTE-V2X	ITS-G5	LTE-V2X	ITS-G5	LTE-V2X
100	MCS 2	MCS 6	MCS 2	MCS 9	MCS 4	MCS 13
200	MCS 2	MCS 9	MCS 2, 4	MCS 9	MCS 4	MCS 13
300	MCS 4	MCS 9	MCS 2	MCS 9	MCS 4	MCS 13
400	MCS 4	MCS 9	MCS 2	MCS 9	MCS 4	MCS 13
500	MCS 4	MCS 9	MCS 4	MCS 9	MCS 4	MCS 13

## Data Availability

Data used to obtain the results presented in this paper can be found online at this address: https://drive.google.com/file/d/1QlsQZnRmNiJDquRE94iTn1TGsKA6FuKM/view?usp=sharing (accessed on 4 December 2022).

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
