# Peer review of "Performance Analysis of Existing ITS Technologies: Evaluation and Coexistence"

_sensors, 2022, doi:10.3390/s22249570_

Round 1

Reviewer 1 Report

An analysis model to associate different constraints related to environment-changing situations is proposed. A coexistence solution between ITS technologies based on density-sharing to better support the requirements of C-ITS services is presented. This study identifies the suitable configurations for each technology, revealing a significant challenge in vehicular networks.  

Some modifications need to be further supplemented and improved: (1) The abstract needs to be further modified and improved to highlight the main innovations and contributions of the paper. (2) The third part, "Performance evaluation and comparison", is the core of the manuscript. It is recommended to further refine and supplement it for detailed description.

Author Response

The point-by-point response to the reviewer's comments is available in the attached file. Thanks to the reviewer for the time he devoted to our article.

Reviewer 2 Report

The authors propose a comprehensive analysis model that can evaluate ITS technologies boundary according to different parameters such as data rate, channel capacity, and density. 

The topic is actual and interesting. The paper appears well written and organized, even if it can be improved in some parts, especially in abstract and Introduction to better clarify the objective and the contribution in a clear way (for example, writing about boundaries is too generic and can be misunderstood). The part related to coexistence and comparison is quite interesting and deserve attention.

I have some cooments.

To make the work more novel, the authors should add a section also on 5G-V2X. Even if authors do not add new results on 5G-V2X, they could however discuss the future behaviour.

The authors use an highway scenario. Do they believe that the model could apply to urban cases?

Can the authors provide more details on simulation settings?

Figures seem to be plotted in excel which is not a good scientific tool for graph. The authors should replot the figures with Matlab o other tools.

The authors should bette justify the behaviour of the curves. For example, in fig. 2 it’s just a problem of simulation length or why the curves behave that way (especially the case 1200)?

Author Response

The point-to-point response to the reviewer's comments are available in the attached file. Thanks to the reviewer for the time he devoted to our article.  

Round 2

Reviewer 2 Report

The authors answered to all the reviewers' comments, improving the quality of the work and also adding new results. They really did a great job. In my opinion, the paper can now be considered for publication in this journal.